# Understanding the Molecular Arrangement and Orientation Characteristics of Mesophase Pitch and Its Fibers via a Polarized Light Microscope

**DOI:** 10.3390/polym16081114

**Published:** 2024-04-16

**Authors:** Jingpan Li, Ximing Tang, Ji Qin, Jianxiao Yang, Xiao Wu, Yuxin Wei, Xubin He, Zujian Huang

**Affiliations:** 1Hunan Province Key Laboratory for Advanced Carbon Materials and Applied Technology, College of Materials Science and Engineering, Hunan University, Changsha 410082, China; jingpan@hnu.edu.cn (J.L.); wu.tang.xi@foxmail.com (X.T.); 13203123728@163.com (J.Q.); wuxiaodarren@163.com (X.W.); 202114010313@hnu.edu.cn (Y.W.); hexubin@hnu.edu.cn (X.H.); huangzujian@hnu.edu.cn (Z.H.); 2Hunan Province Engineering Research Center for High Performance Pitch-Based Carbon Materials, Hunan Toyi Carbon Material Technology Co., Ltd., Changsha 410221, China

**Keywords:** mesophase pitch, carbon fibers, polarized light microscope, optical texture, microstructure

## Abstract

A polarized light microscope (PLM) was utilized to examine the optical textures of mesophase pitch (MP) and MP-derived fibers, which aimed to reveal the arrangement and orientation characteristics of pitch molecules and to clarify the evolution and transformation mechanism of carbonaceous microcrystalline from pitch fibers to graphitized fibers. The results found that there were distinct optical textures in MP, where one side exhibited a transition from a flattening plane to a mountain-like undulating plane. This transition corresponded to the arrangement of pitch molecules, resembling stacked lamellar structures reminiscent of curved paper. Meanwhile, the optical textures of fibers revealed that the blue substance was wrapped around the red grain-like domains in the longitudinal section and confirmed that the red part belonged to the pyridine insoluble fraction of MP and the blue part belonged to its pyridine-soluble fraction. After graphitization, the red part was transformed into graphite sheets and the blue part was transformed into an amorphous carbon layer which was wrapped around the graphite sheets, forming a carbonaceous microcrystalline package-like bag. Therefore, this study provided a comprehensive interpretation of the structural evolution mechanism of MP and MP-derived fibers based on their macro-optical textures and micro-nanostructures.

## 1. Introduction

Mesophase pitch (MP) serves as an excellent precursor for producing high-performance carbon materials, including carbon foam [1], needle coke [2], and carbon fibers (CFs) [3]. Especially, MP-based CFs (MPCFs) exhibit exceptional mechanical, electrical, and thermal properties and are widely utilized in aerospace and military applications [4,5,6,7]. Characterizing the structures and properties of MP and MP-derived fibers plays a significant role in its quality control, process optimization, and applications of MPCFs. In the past few decades, many approaches have been used to attempt to analyze the molecular structure of MP and the microcrystalline structure of its fibers, aiming to establish the relationship between the structure and properties of MPCFs. For example, the molecular structure model of MP was speculated based on solid ^13^C-NMR and ^1^H-NMR technology [8]. The molecular weight and distribution of MP was evaluated through the advantage (MALDI TOF-MS) method [9,10]. The aggregation structure of MP and the microcrystalline structure of MPCFs were observed using the (XRD) and Raman spectra [11,12]. These ^13^C-NMR, ^1^H-NMR, MALDI TOF-MS, XRD, and Raman characterization approaches could reveal the molecular structure and microstructure of MP and MP-derived fibers, but they were limited by understanding of their molecular arrangement and orientation in three-dimensional space. Actually, the optical texture of MP has been extensively studied using polarized light microscopy (PLM) even with a hot stage. Brooks and Taylor [13] firstly observed that there were many anisotropic spheres in the synthetic process of MP. These spheres subsequently grow, aggregate, and form the bulk mesophase. Zimmer and White [14] also investigated the polarization structures of MP using PLM and scanning electron microscopy (SEM). They proposed several sheet models to depict the structural representation of the carbonaceous mesophase, highlighting diverse polarized structures within MP that result in distinct colors when observed under PLM images. Based on these findings, they established a correlation between the molecular arrangement of MP. Chen [15] observed the microstructure of carbonized MP using SEM and identified distinct curved carbon layers, aligning with Zimmer′s proposed model. Moreover, Amod A. Ogale [16,17,18,19] further reported the dynamic rheology and accompanying microstructure of MP during the shear-flowing process based on the PLM micrographs of MP samples for the initial state and after steady shearing, as observed in three orthogonal sections. They pointed out that the initial microstructure, examined in three orthogonal sections, was found to have a weak, but preferential, orientation of mesophase layers in the radial direction of the rheometer plate, and the initial microstructure changed to a flow-aligned fibrous structure after shearing flow. In addition, Mochida [20] examined MP under the extraction treatment with solvents like tetrahydrofuran (THF) and pyridine. SEM analysis revealed rod-shaped microdomains within the MP structure, demonstrating their potential for extrusion into fibers with diameters in the tens-of-microns range through melt-spinning techniques. Gerald [21] discovered that MP-derived fibers resembled spherical or elliptical grains on their transverse surface, while thin needle-like or ribbon-like domains were present in the longitudinal sections of MP-derived fibers, as observed through PLM analysis. Mochida [22] further investigated the domain structure of MPCFs on cross-sections and categorized them into linear, curved, and ring-like domains. Hong [23] identified similarities between the microdomains within the extracted spun fibers and the insoluble fraction of MP. Additionally, folded units were observed on the longitudinal surface of carbonized and graphitized fibers. The development of microdomains in MP-derived fibers throughout different processes was also examined, revealing significant changes in their shape and size after the heat treatment. Moreover, during the SEM examination of the skin area of MPCFs, Hong [24] observed the presence of bright spurs aligned along the longer axis of the fibers. Following the treatment of fibers with pyridine, the basal plane comprising small molecules vanished, revealing direct exposure of graphene edges on the fiber surface.

Despite numerous prior investigations focusing on the structure of MP and its CFs, there remain unresolved issues that require further elucidation. Previous studies investigating the optical texture of MP molecules using PLM have been primarily focused on by observing MP from a single side, leading to a limited understanding of the overall arrangement and orientation characteristics of MP molecules in 3D space. To address these limitations, this study aims to observe the three-dimensional surfaces of MP via PLM to and establish a 3D polarized texture model for MP. In addition, the longitudinal PLM images of MPCFs showed numerous elliptical structures, which will be referred to as microdomains. However, most of the existing literature describing these microstructures of MPCFs relied on their radial cross-section SEM images, where the presence of elliptical microdomains were not visible. This discrepancy calls for an integrated model to better understand both the macro-optical texture from PLM images and micro-nanostructure from SEM images of MP and its fibers. Hence, this work aimed to investigate the structural characteristics of MP and MP-derived fibers during the stabilization, carbonization, and graphitization processes via PLM and SEM. Ultimately, it was expected to provide a comprehensive understanding of the macroscopic optical texture and microstructure evolution mechanisms present in both MP and MP-derived fibers.

## 2. Materials and Methods

### 2.1. Materials

MP with a softening point of 275 °C was prepared using the FCC-DO as a raw material, which was supplied by Toyi Carbon Material Technology Co., Ltd., Changsha, China.

### 2.2. Preparation of MP-Derived Fibers

MP-derived pitch fibers (PFs) were prepared by the melt-spinning method with a single-hole of spinneret (L/D = 0.4 mm/0.2 mm) at a spinning temperature of 355 °C and a nitrogen pressure of 0.6 MPa. Then, the PFs were stabilized at 270 °C for 1 h with a heating rate of 1 °C/min in a 200 mL/min air atmosphere to obtain stabilized fibers (SFs). The SFs were subsequently carbonized at 1000 °C for 1 h with a heating rate of 5 °C/min in a 200 mL/min nitrogen atmosphere to obtain the carbonized fibers (CFs) and were further graphitized at 3000 °C for 10 min in a graphitization furnace to obtain the graphitized fibers (GFs). Moreover, the PFs were extracted by the Soxhlet extraction method with the pyridine solvent for 12 h; the pyridine-extracted PFs are denoted as P-PFs for comparison.

### 2.3. Characterization of MP and MP-Derived Fibers

MP-observation sample preparation: firstly, the MP block was fitted into the fold of tin foil with a three-dimensional shape. Then, it was put into the cylindrical rubber mold, adding the epoxy resin with a few drops of curing agent (triethylene tetramine). When it was solidified, the cylinder with MP was obtained after releasing it from the mold. Finally, the cylinder was carefully hand-polished to a bright mirror finish on the three-dimensional sides with 100, 400, 800, and 1000 polishing papers in turn, and it was at last polished with the polishing cloth, adding some polishing agents (alumina dispersion, <0.5 μm) to obtain an observed MP sample on the three-dimensional sides.

Observable MP-derived-fiber-sample preparation: A bundle of MP-derived fibers was placed on the bottom of the cylindrical rubber mold, adding the epoxy resin with a few drops of curing agent (triethylene tetramine). When it was solidified, the cylinder with the MP-derived fibers was obtained after releasing it from the mold. Finally, the cylinder was carefully hand-polished to a bright mirror finish on the three-dimensional sides with 100, 400, 800, and 1000 polishing papers in turn, and it was at last polished with a polishing cloth, adding some polishing agents (alumina dispersion, <0.5 μm) to obtain observable MP-derived-fiber samples in the direction of the longitudinal section of fibers.

The optical texture of MP and MP-derived fibers were observed using a BX53 polarized light microscope (PLM, Olympus, Tokyo, Japan) with the prepared observable MP samples in the orthogonal mode.

The microcrystalline structure of MP-derived fibers was analyzed using a DXR2 Raman spectrum (Thermo Fisher, Waltham, MA, USA) at an acceleration voltage of 5 kV and a wavelength of 563 nm with the observed MP-derived fibers samples in different positions.

The morphology and microstructure of CFs and GFs were analyzed using a SU8010 scanning electron microscope (SEM, Hitachi, Tokyo, Japan) at 5 kV with the brittle fracture cross-section of fibers under the liquid nitrogen condition.

## 3. Results and Discussion

### 3.1. The Optical Textures of MP

Figure 1 presents the PLM images of MP in the three sides, namely the A-, B-, and C-sides. In Figure 1b, a significant area of the pitch, is observed to be peeling off with multiple cracks present. Additionally, the characteristic streamlined orientation commonly seen in PLM images of MP is not evident on this side. Instead, homochromatic domains are embedded on the surface, resembling a covering. Further grinding and polishing of this side leads to the PLM images in Figure 1c, revealing raised continuous mountain-range-like structures. It is worth noting that this phenomenon is repeatable.

In contrast, the B-side and C-side of MP do not exhibit non-lamellar exfoliation after the polishing treatment, and their PLM images remain unchanged, demonstrating the streamlined texture characteristic of MP as shown in Figure 1d,e. Furtherly, Figure 1f,g highlights a more pronounced stripe feature, with the yellow arrow indicating the direction of the polarization stripe and the green dashed line denoting the junction between the B-side and C-side. Overall, the polarized texture of the B-side and C-side depicted in Figure 1 exhibits a symmetric relationship, which is consistently observed.

Additionally, in this experiment, the normal direction of the MP molecular plane in the B-side and C-side was not completely perpendicular to the plane of the polishing cloth, which made the macrostructure of MP in the B-side and C-side not to easily changeable under the limited polishing treatment. As for the A-side, the normal direction of the MP molecular plane was perpendicular to the plane where the polishing cloth was located, and the van der Waals force between the molecular layers of MP on the A-side was easily destroyed by the shearing forces during the polishing treatment. Therefore, the molecular layer of MP on the A-side is more easily damaged or shed; that is, it is easier to expose the arrangement and orientation of the molecular structure characteristics of MP on the A-side. In fact, by continuing to polish the MP on the B-side and C-side until the fold structure is exposed, the PLM images of MP on the B-side and C-side are also likely to be changed.

Regarding the relationship between the arrangement pattern of MP and PLM images, Zimmer [14] proposed the model illustrated in Figure 2. This model suggests that the bent arrangement of MP molecules exhibits different polarization features depending on the viewing angle. When observing in the direction of the bent molecule’s edge, two interference bands with distinct colors can be observed from the surroundings. However, when viewing in the direction of the side of the bent molecule, only one interference band with different colors from the surroundings is visible. By considering this model along with the observed polarizing stripes on the C-side combined with those on the B-side, it is indicated that the molecular structure of MP is similarly oriented to stacked paper.

Combining the PLM images of MP on the B-side and C-side, as shown in Figure 1d–g, it can be found that there are two yellow stripes on one side and that one stripe on the other side is perpendicular to them, which is exactly in the middle of them. The fringe arrangement characteristics of MP in the junction PLM images of the B-side and C-side under the orthogonal mode are in accordance with Zimmer’s results and description. Moreover, the trend of the yellow stripes also indicates the normal direction of the maximum curvature in the fold structure to some extent. It is noticed that there are some cracks on the C-side, which may be caused by the fact that the normal direction of the molecular arrangement on the top of the fold structure of MP on the C-side is not consistent with the polishing cloth during the polishing treatment. This feature further confirms the reason why the molecular layer of MP on the A-side easily falls off. Meanwhile, it can be found that there is no this feature on the B-side, because the normal direction of the molecular structure of MP on the B-side is coplanar with the polishing cloth. Consequently, by integrating the experimentally obtained stereoscopic PLM images of MP and Zimmer’s proposed model [14], the 3D polarizing model of MP is depicted in Figure 3. This model illustrates that the arrangement of MP molecules undergoes a transition from a flat surface on one side to a mountainous undulating surface on the other side. This transition corresponds to the arrangement of MP molecules resembling folded paper stacked beneath flat stacked molecules. The orientation of the “folded paper” may vary, resulting in different polarization textures on different sides. Therefore, compared with previous work, the innovation of the proposed model lies in the discovery of the distribution and orientation of the fold structure in the MP block through the macroscopic phenomenon and microscopic observation of its PLM images.

### 3.2. The Optical Textures of MP-Derived Fibers

According to the previously published works of Mochida [24] and the PLM image model of MP-derived PFs as shown in Figure 4, the red grain-like structure observed in MP-derived PFs is referred to as a “domain”. This domain primarily comprises the pyridine-insoluble (PI) fraction of MP. On the other hand, the blue component represents a substance that fills the gaps within the domains, which may correspond to the pyridine-soluble (PS) fraction of MP.

To confirm the composition of the red and blue components, the MP-derived PFs were subjected to Soxhlet extraction using pyridine solvent. The PLM images of pyridine-extracted MP-derived PFs before and after pyridine solvent extraction are presented in Figure 5. These images clearly demonstrate a significant reduction in the blue component at the skin of the pyridine-extracted fibers, with noticeable break-downs being evident. Consequently, it can be concluded that the blue component corresponds to the PS fraction of MP, while the red component originates from the PI fraction of MP.

Moreover, distinct differences were observed between the core and skin areas in the PLM images of SFs, as depicted in Figure 6. In the core of MP-derived SFs, both in longitudinal and cross-sectional directions, the fusion of domain structures was evident, resulting in a more irregular polarized texture. Conversely, the skin of MP-derived SFs retained the characteristic red fruit-like tissue, with no significant variation in the polarized features compared to the original PFs. These structural features originate from the stabilization stage, where inconsistencies arise between the core and skin structures due to the limited penetration of oxygen into the core areas of the fibers, preventing the formation of an oxidized cross-linked structure. The aforementioned characteristics directly exemplify this phenomenon via PLM images, illustrating that the core region undergoes re-melting at the stabilization temperature due to the absence of an oxidation cross-linked structure. This phenomenon persists and is reflected in the carbonized and graphitized fibers.

After carbonization, rectangular domains were observed when the cross-section of CFs was parallel to the graphite sheets, while longer grain-like domains were observed when the cross-section was perpendicular to the graphite sheets. This can be attributed to the removal of oxygen and the gradual connection of carbonaceous microcrystalline, leading to the formation of graphite sheets during the carbonization process. Similarly, larger graphite sheets were observed in GFs when the cross-section was parallel to the graphite sheets, while longer grain-like domains were observed when the cross-section was perpendicular to the graphite sheets. Additionally, interconnected domains were also observed on the skin of CFs and GFs when the image sections were neither parallel nor perpendicular to the graphite sheets, as shown in Figure 7. Overall, the development of the polarized texture characteristics aligns with previous theories. Grain-like domains are observable in PFs, and, after stabilization, the domains in the core area of SFs can melt and come into contact with each other. Whereafter, the size of domains can increase after the carbonization and graphitization processes, leading to different polarized textures in various cross-section PLM images of CFs and GFs.

### 3.3. The Microcrystalline Structure of MP-Derived Fibers

In order to explore the microcrystalline structure evolution of MP-derived fibers during the stabilization, carbonization, and graphitization processes, the Raman spectra of MP-derived fibers are shown in Figure 8. The Raman D-band, at around 1360 cm^−1^, and the G-band, at around 1580 cm^−1^, correspond with the defect lattice-vibration mode and ideal graphite lattice-vibration mode, respectively. Therefore, the intensity ratio of the D-band and G-band (I_D_/I_G_) could be used to evaluate the carbon crystallite character, and a lower value of I_D_/I_G_ indicates fewer defect sites, while a more highly ordered orientation of fibers [25]. It was noticed that the I_D_/I_G_ of MP-derived fibers were subsequently increased from PFs to SFs and CFs, and the I_D_/I_G_ of GFs were of a minimum, were related to the oxygen cross-linked structure of SFs, and some defect structures of CFs and highly ordered carbon crystallite orientations of GFs. These results were in accordance with the polarizing texture evolution of MP-derived fibers at various stages. Moreover, in order to reveal the observed differences between the red region and the blue region of the fruit-grain domain structure in the PLM images of MP-derived fibers, the molecular arrangement characteristic of MP-derived fibers in the corresponding core and edge positions of fruit-grain domain structures were carefully analyzed using Raman spectra, as shown in Figure 8. It was noticed that there was a small difference in the I_D_/I_G_ of PFs between the core (red region) and the edge (blue region). After stabilization, carbonization, and graphitization, the I_D_/I_G_ difference of MP-derived fibers in the core and edge positions of fruit-grain domain structures was gradually increased, especially in GFs (GF—core = 0.43, GF—edge = 0.79). This phenomenon was due to the gradual fusion of the granular structure (core position), which is mainly composed of PI fractions of MP (red part in PLM images), forming a more regular graphite sheet during the heat-treatment processes and appearing at a high graphitization degree. Meanwhile, the substrate (edge position) composed of PS fractions of MP (blue part in the PLM images) was transformed into an amorphous carbon layer, showing a low graphitization degree. These observations were consistent with our proposed conclusions from the PLM images of MP-derived fibers.

### 3.4. The Microstructure of MP-Derived GFs

The various distinctive radial structures of MP-derived GFs were observed in the cross-section SEM images, as shown in Figure 9. Notably, package-like microstructures were apparent in Figure 9a–c. It is important to note that these GFs are observed after being cut, thus revealing noticeable gaps at the top of the “packages”. Curved edges are observed in Figure 9d,e, when the cutting surface is just below the top of the “packages”. Furthermore, in Figure 9f, a large oval edge with black graphite sheets embedded within it can be observed when the cutting surface is positioned nearly in the middle of the “packages”. Additionally, boat-like structures are clearly visible in Figure 9g–i, which are thought to represent the bottom of the “packages”.

In line with previous theories, the bright regions observed in the SEM images of GFs are associated with the PS fraction of MP [20]. Furthermore, Figure 9 clearly illustrated that the morphology of the GFs on the SEM images is derived from the PS fraction of MP after graphitization and resembles a package, and the observed structure varies depending on the cutting position of the GFs.

### 3.5. The Interpretation of Structural Evolution between the Optical Structure and Microstructure of MP-Derived Fibers

Based on the above analysis, the grain-like domains are observed in large numbers and appear repeatedly in PLM images (Figure 6). And these domains eventually transform into a package-like structure in SEM images (Figure 9) during the structural evolution of MP-derived fibers. To explain this phenomenon, two research methods are commonly used. The first method involves studying it from an overall perspective, as demonstrated in Figure 6. The second method involves studying the specific shape of these domains by isolating a structural unit. Moreover, this structural unit should include an elliptical red domain and the surrounding blue region. That is, the red domain represents the PI fraction, while the blue region is the PS fraction which wraps around the PI fraction even after carbonization and graphitization. In other words, the assumption is that if the domain structures from the PI fraction do not interconnect before carbonization, the PS faction would encase the elliptical domain structure, forming a larger ellipse.

Thus, the models of the structural unit “package” and domain of MP-derived fibers are depicted in Figure 10. After graphitization, the red domain structures gradually form a graphite flake structure in a specific manner, which is evident in the SEM images and corresponds to the content presented in the upper half of Figure 10. Regarding the horizontally interconnected structural units after graphitization, when observed using a PLM at an angle parallel to the graphite flakes, larger flake-like graphite structures are observed. When observed from an angle perpendicular to the graphite flakes, longer grain-like domain structures are observed. These structural characteristics are also reflected in SEM images, as described in Figure 9. If a cross-section is taken from the midpoint of the height of the GFs, it can be observed that the graphite structure is enveloped by the amorphous carbon layer, and the cross-sectional shape matches the proposed cross-sectional diagram of the structural unit in Figure 10. If the cross-section is slightly above or below the midpoint, the edges of the section will exhibit curved features. If only a small piece from the bottom is retained through shearing, it will resemble the shape of a boat, according to the model of the structural unit.

## 4. Conclusions

The 3D PLM images of MP revealed that a transition from a flattening plane to a mountain-like undulating plane on one side could be observed. However, on the other two sides, the polarizing textures appeared symmetric, resembling structures similar to stacks of folded paper. Consequently, different PLM images were obtained when observing MP from different sides. The optical texture of MP-derived fibers could be observed in the longitudinal section of fibers from PLM images, showing the red fruit-grain domain structure and the blue substance wrapped around the domain structure. After carbonization and graphitization, the red domain structure would become fused and mature. That is, in PLM images of PFs, grain-like domains composed of PI fractions were observed, which underwent size and shape changes during carbonization and graphitization processes. The graphitized portions derived from the PS fraction formed package-like structures after graphitization, and SEM observations revealed special features, such as gaps, curved edges, and the bottoms of the packages within the cross-section images of GFs. A series of models was presented to provide a unified interpretation, integrating the PLM and SEM characteristics of fibers at all stages. Ultimately, these models have the potential to deepen the understanding of the structural units of MP and its fibers. Therefore, this work successfully gave a unified explanation of the evolution mechanism of MP and its fibers between the optical texture and microstructure, which can be expected to provide a certain theoretical foundation for the regulation and performance optimization of MPCFs.

## Figures and Tables

**Figure 1 polymers-16-01114-f001:**
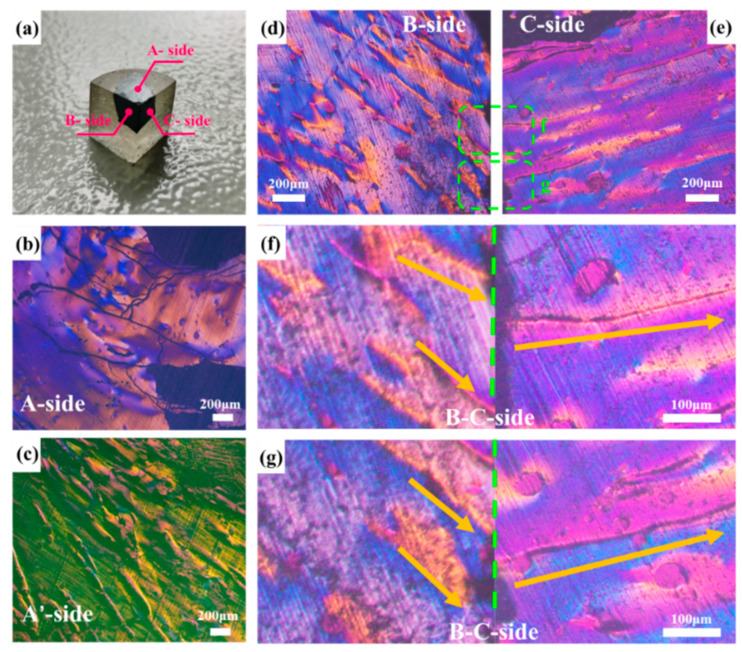
Physical sample of MP (**a**), PLM images of MP on an A-side (**b**), A’-side (**c**), B-side (**d**), C-side (**e**), enlarged view of B-C side (**f**,**g**). The arrow represents the orientation direction of the polarized structure and the green dashed line represents the intersection line of the B-C side.

**Figure 2 polymers-16-01114-f002:**
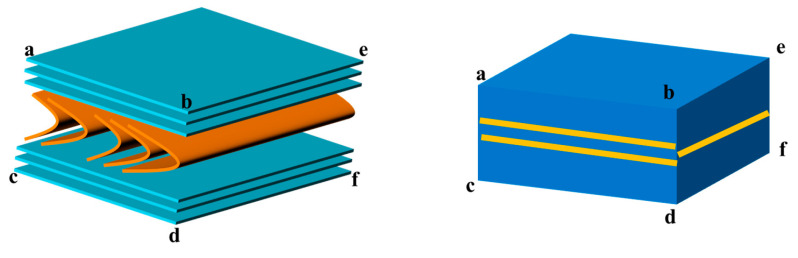
A relationship between polarized stripes and molecular orientation of MP proposed by Zimmer [14]. The yellow lines represent the polarization features of the corresponding molecular orientation, and the letters (a,b,c,d,e,f) represent the corresponding observation direction plane.

**Figure 3 polymers-16-01114-f003:**
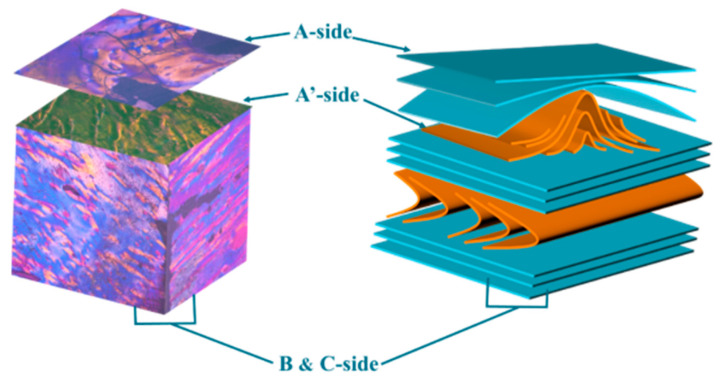
3D PLM images’ model and molecular structure arrangement, and an orientation model of MP. The blue component of the model represents the laminar structure of MP molecule, while the orange component represents the folded structure of MP molecule.

**Figure 4 polymers-16-01114-f004:**
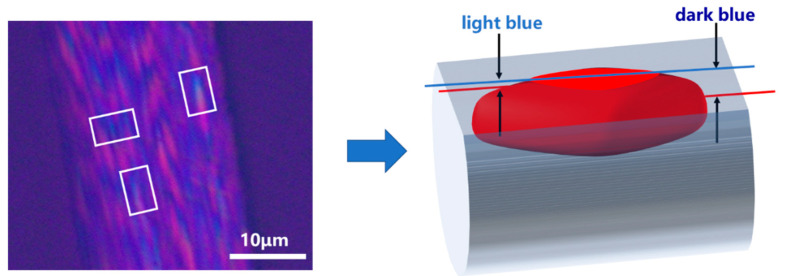
The PLM image model of MP-derived PFs based on the literature of Mochida [24]. The model is derived from the characteristic in the light blue area of PLM images (white box).

**Figure 5 polymers-16-01114-f005:**
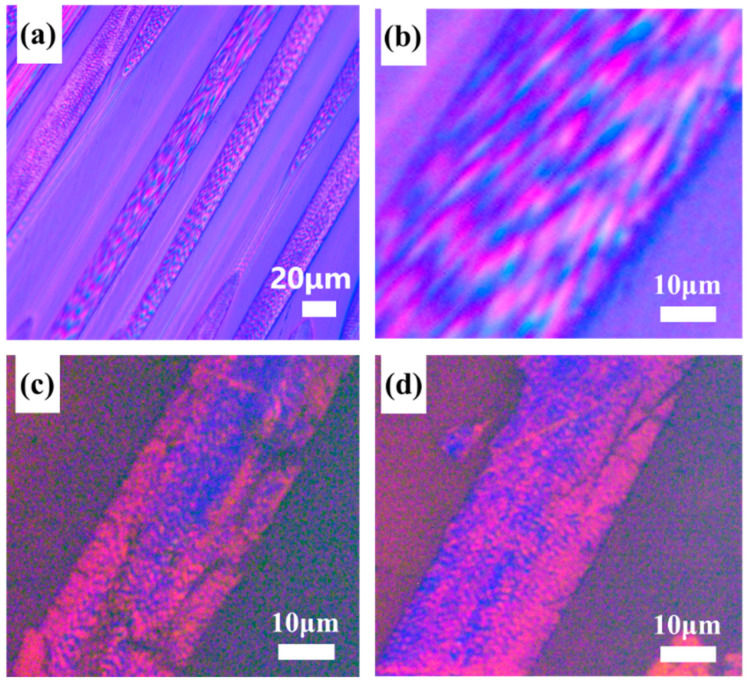
PLM images of pyridine extracted MP-derived PFs (**a**,**b**) before extraction and (**c**,**d**) after extraction.

**Figure 6 polymers-16-01114-f006:**
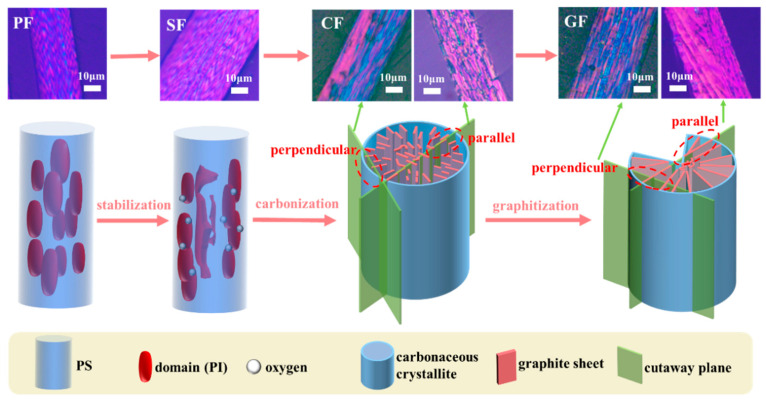
The evolution model of polarizing textures of MP-derived fibers at various stages.

**Figure 7 polymers-16-01114-f007:**
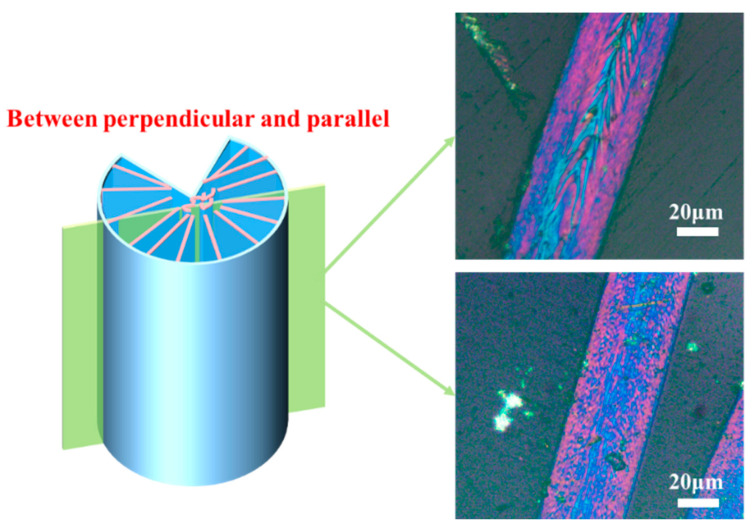
PLM images of CFs and GFs in the cross-section between perpendicular and parallel.

**Figure 8 polymers-16-01114-f008:**
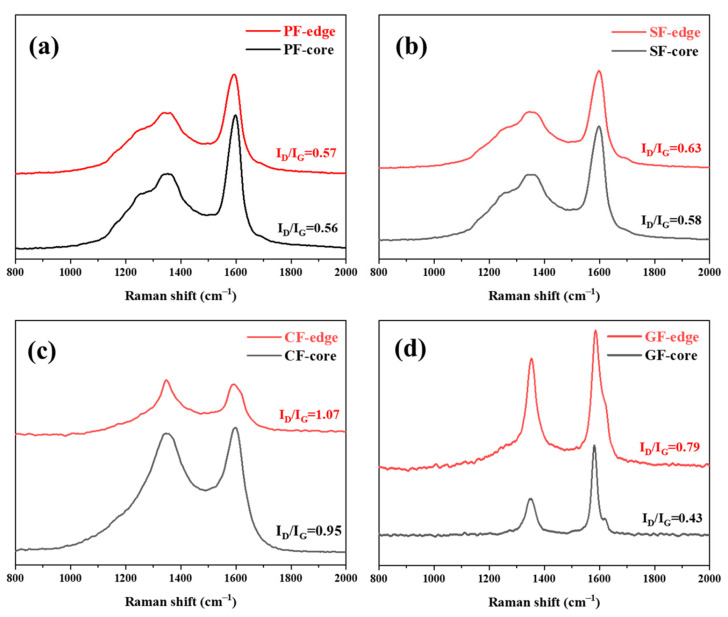
Raman spectra of MP-derived PFs (**a**), SFs (**b**), CFs (**c**), GFs (**d**) in the core (red part) and edge (bule part) positions of fruit-grain domain structures.

**Figure 9 polymers-16-01114-f009:**
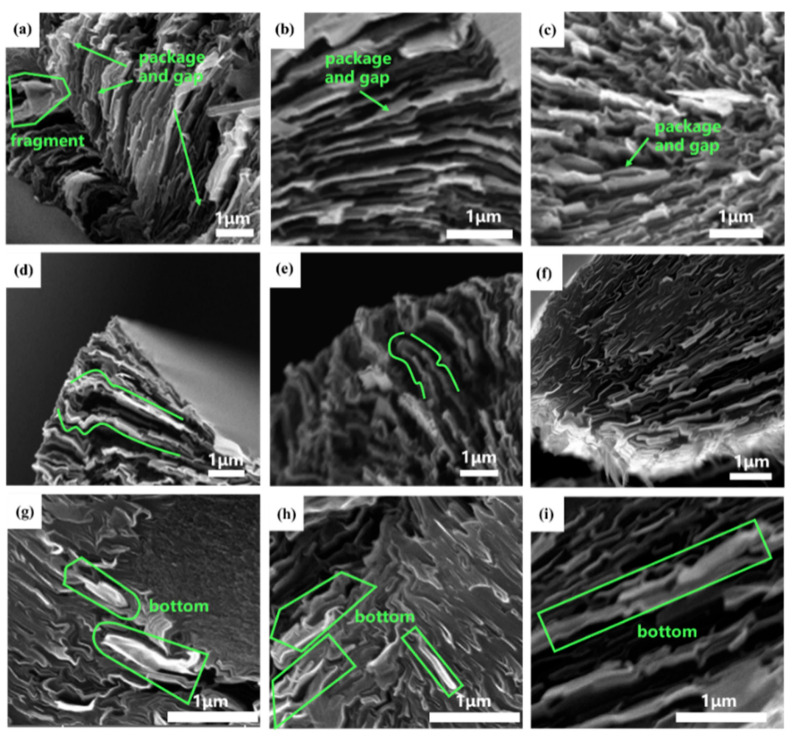
The cross-section SEM images of MP-derived GFs. The top opening part (**a**–**c**), the middle part (**d**–**f**) and the bottom part (**g**–**i**) of the packages structure. The green box area shows the typical an amorphous carbon layer around the graphite sheets of GFs.

**Figure 10 polymers-16-01114-f010:**
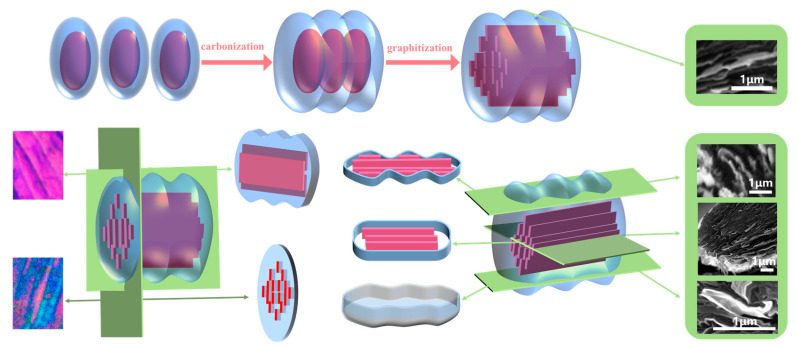
The models of “package” and domain of MP-derived fibers.

## Data Availability

Data are contained within the article.

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
