# Peer review of "Understanding the Molecular Arrangement and Orientation Characteristics of Mesophase Pitch and Its Fibers via a Polarized Light Microscope"

_polymers, 2024, doi:10.3390/polym16081114_

Round 1

Reviewer 1 Report

Comments and Suggestions for Authors

Ref.comments to the paper titled as “Understanding the molecular arrangement and orientation characteristics of mesophase pitch and its fibers via the polarized light microscope” written by the authors: Jingpan Li, Ximing Tang, Ji Qin, Jianxiao Yang, Xiao Wu, Yuxin Wei, Xubin He, Zujian Huang.

It is well known that the study of the molecular arrangement and orientation characteristics, etc. is useful for the general physical knowledge about the materials features and for the practical application of some optical devices, for example, polarized microscope. From this point of view, the current paper is modern and actual.

For the first, this paper is included the analysis of 19 literature data. Indeed, the authors are known the problem and can make some steps to resolve the important tasks. But it is not enough for this perspective area of the study. Please add in your analysis at list 5-7 papers written by last 3-5 years. You can show how different approaches can be used to study the molecular arrangement, in comparison with the microscope method.

Well. The paper is good illustrated; the pictures shown can help to the reader to understand the article text with good advantage.

Preparation of the structures studied and the instrumentations (Microscope with the polarized light, SEM-device) used for the experiments is good.

Please explain which kind of the substrate has been used to put your structure inside the microscope? It is important due to the reason that the surface of the substrates (polished, unpolished, glass, quartz, silicon-daces substrates, etc.) is influence on the wettability, thus it is influence on the relief used to obtain the molecular arrangement and orientation.

Results and discussion part. Physical sample images, their simulations, SEM-images – all are the best. But, please show the spectra of your structure. It is necessary to add your data via spectral analysis.

The paragraph of “The interpretation of structural evolution between the optical structure and microstructure of MP-derived fibers” is interesting. The explanation is not in contradiction in our basic physical knowledge. Once again, this paragraph can be extended via spectral parameters adding. It permits to extend our vision about what kind of the transformation you can observed in your structures.

Conclusion part is so short; it should be extended. It is not included the basic results presented in the current paper.

As for my local opinion, the paper can be published after major corrections.

Author Response

Thank you very much for the recognition of our work. We have tried our best to revise our manuscript based on your valuable constructive comments.

Please review the "Response to Reviwers".

Reviewer 2 Report

Comments and Suggestions for Authors

This manuscript discussed the utilization of polarized light microscope (PLM) to examine the optical textures of mesophase pitch (MP) and MP-derived fibers. This investigation aimed to elucidate the arrangement and orientation characteristics of pitch molecules and to clarify the evolution and transformation mechanism of carbonaceous microcrystalline structures from pitch fibers to graphitized fibers. Different PLM images were obtained when observing MP from different sides. There are some issues that should be addressed before publication.

1.     In Page 3, line 117, Please explain why the PLM images of B-side and C-side of MP remained unchanged after the polishing treatment?

2.     In Page 4,line 128, Discussing the model proposed by Zimmer [9] briefly is appropriate, however, too much discussion on the work of other researchers might resemble literature review.

Author Response

Thank you for acknowledging our efforts. We have diligently revised our manuscript in response to your valuable constructive comments.

Please review the "Response to Reviwers".

Reviewer 3 Report

Comments and Suggestions for Authors

I have revised the paper etitled: "Understanding the molecular arrangement and orientation 2 characteristics of mesophase pitch and its fibers via the polarized light microscope" and these are my comments and suggestions:

 - The figures in the paper should be much bigger in oreder to be easier to read them. 

- Refferences must be newer than 10 years. You no have 13/19 references older than 10 years, so they can be cosidered outdated.

- Chapters 3.1 and 3.2 need to be discussed in more detail than there are now since you only assessed the structure of the polarized textures (3.1) and obtained MF (3.2) and just concluded that they are similar as described by the literature (Zimmer et. al. [9] for chapter 3.1 and Mochida et. al. [19] for chapter 3.2).

- chapter 3.3 needs also a deeper discussion since you only presented the cross-section of the GF and you made no interpretation on why this arrangemnt occured.

Overall, the paper seems more like a technical report in which you obtained GF for mesophase pitch, you observed the structure of the pitch and cross-section of GF and concluded that the transformation are as the prevoius papers (more than 10 years old ones) predicted. The scientific insight is very low and no discussion was made to assess why the phenomenon occur and why they occur as they do.

Author Response

(The authors gave the same response as above.)

Round 2

Reviewer 1 Report

Comments and Suggestions for Authors

Second step of the ref.comments to the paper titled as “Understanding the molecular arrangement and orientation characteristics of mesophase pitch and its fibers via the polarized light microscope” written by the authors: Jingpan Li, Ximing Tang, Ji Qin, Jianxiao Yang, Xiao Wu, Yuxin Wei, Xubin He, Zujian Huang.

I have seen the revised version of the current paper. As for my local opinion the paper now is dramatically improved for the reader’s understanding. Really, Introduction section, Materials and Methods section, Results and Discussion one are extended drastically. Moreover, Conclusion part is extended as well. Furthermore, the additional analysis of the papers (added in the reference list) has been made.

Thus, the paper can be published in the current form!

Reviewer 3 Report

Comments and Suggestions for Authors

Dear authors, 

Thank you for taking into consideration the comments and suggestions I made regarding your paper.

Now, is all better and I consider it is enough for publishing.